# The magnitude and associated factors of immediate postpartum anemia among women who gave birth in east Gojjam zone hospitals, northwest- Ethiopia, 2020

**Getachew Altaseb Agmassie[1], Genet Degu Alamneh[2,3], Moges Wubie Ayicheh[2,3], Worku Taye Getahun[1]\*, Aysheshim Asnake Abneh[1]**

1 Department of Midwifery, Debre Markos Comprehensive Specialized Hospital, Debre Markos, Ethiopia, 2 Departments of Midwifery, Debre Markos University, Debre Markos, Ethiopia, 3 Departments of Public Health, Debre Markos University, Debre Markos, Ethiopia

\* worku384@gmail.com

**Data Availability Statement:** All relevant data are within the paper and its Supporting Information files.

## Abstract

### Background

Globally around half a million maternal death occurred annually related to labor and delivery of which twenty percent is contributed by post-partum anemia. Postpartum anemia contributes about two percent of total maternal mortality in Ethiopia. Immediate postpartum anemia is a common public health problem in most parts of the globe, being frequent in low and middle-income countries including in the developed world. The previous studies cut off point for immediate postpartum Anemia is 11mg/dl which is the cutoff point of anemia after one week of postpartum, environmental factors like barefoot were not addressed in the previous studies and the previous studies were conducted in a single facility This study aimed to assess the magnitude and associated factors of immediate post-partum anemia among women who gave birth in East Gojjam zone hospitals, Northwest Ethiopia.

### Methods

Institutional based cross-sectional study was conducted from October 20-November 20 2020 on immediate post-partum anemia. During the study 467 study participants were included by using systematic random sampling method Data were collected using a structured interviewer-administered questionnaire and a blood sample was used for hemoglobin determination. Data were checked, coded, and entered into Epi-Data Version 4.2 and then exported to SPSS version 25 for analysis. Binary logistic regressions were done to identify predictors of immediate post-partum anemia and a 95% confidence interval of odds ratio at a p-value less than was taken as a significance level.

### Results

The overall magnitude of immediate postpartum anemia among mothers who gave birth in East Gojjam Zone Hospitals were found to be 21.63% (95% CI:18.12%, 25.11%), not having antenatal care follow-up (Adjusted Odds Ratio (AOR) = 2.92;95% CI:1.20,7.06), assisted

**Funding:** The author(s) received no specific funding for this work

**Competing interests:** The authors have declared that no competing interests exist.

**Abbreviations:** AOR, Adjusted Odds Ratio; ANC, Antenatal Care; BSC, Bachelor Degree of Science; COR, Crude Odds Ratio; CSAE, Central Esthetical Agency of Ethiopia; DMU, Debre Markos University; EDHS, Ethiopian Demography Health Survey; ERC, Ethical Review Committee, g/dl-Gram Per Deciliter; Hct, Hematocrit; Hgb, Hemoglobin; IFA, Irion-folic Acid; IPPA, Immediate postpartum Anemia; MUAC, Midupper Arm Circumference; NGO, Nongovernmental Organization; SVD, Spontaneous Vaginal delivery, PA-Postpartum Anemia; WHO, World Health Organization.

instrumental delivery (AOR = 2.72; 95% CI:1.08,6.78),mid-upper arm circumferences less than 23cm (AOR = 5.75;95% CI:3.38, 9.79), antepartum hemorrhage (AOR = 4.51; 95% CI:2.42, 8.37), never wearing shoes (AOR = 2.60; 95% CI:1.10, 6.14) were found to be significantly associated with immediate postpartum anemia.

## Conclusion

This study indicates that immediate postpartum anemia is a moderate public health problem in the study area. A more careful strategy is ideal to increase antenatal care follow-up that sticks to national guideline contact schedule, safe reduction of instrumental and cesarean deliveries to the minimum, quick and timely linkage and treatment of malnourished pregnant mothers to the center where they get adequate health care services, along with a high index of suspicion in mothers diagnosed with antepartum hemorrhage, wise and vigilant advice on consistent use of the shoe for pregnant mothers are recommended to tackle the burden of immediate post-partum anemia.

## Introduction

The postpartum period is the most climacteric time for both lives of the women and their newborns. Despite there being high maternal and newborn mortality and morbidity during this time, this is the most forgotten area for the provision of quality health care service, particularly in developing countries including Ethiopia. Anemia is a main global public health problem that affects the health quality of life and working ability of most people all over the world [1]. According to WHO and United kingdom guidelines, Immediate post-partum anemia is defined as a hemoglobin level less or equal to 10g/dl in the immediate puerperium [2, 3].

Globally around 500,000 maternal deaths occurred annually related to labor and delivery of which 20% are contributed by post-partum anemia [4–6]. Postpartum anemia contributes to about 2% of total maternal mortality in Ethiopia [7]. The magnitude of post-partum anemia in the developed country varies from 10–30% and in low and middle-income higher contributes from 50 to 80% [8]. Post-partum anemia is the leading indirect cause of maternal mortality in Ethiopia. According to EDHS 2016 Post-partum anemia account about 29% [9, 10].

Post-partum anemia can be classified based on its public health importance 4.9% and below is normal in prevalence,5–19.9% mild,20–39% moderate, and 40% and above is classified under severe prevalence [11]. Immediate postpartum anemia (IPPA) is a common public health problem in most parts of the globe, being frequent in low and middle-income countries including in the developed world [1].

Immediate Post-Partum anemia (IPPA) hurts the quality of life and well-being both for the mother, and her child. There is the physical contribution of Post-Partum anemia to mothers particularly fatigue, decreased day-to-day working activity, difficulty in breathing during exercise, increased heart rate, dizziness, and increased frequency of infections [8]. Immediate Post-Partum anemia also affects mother-to-child bonding on lactation and their interaction [5].

Although anemia in pregnant women is adequately explained very little attention has been given to immediate postpartum anemia and care during an immediate postpartum period is the neglected part of maternal care [12]. According to the previous studies cut off point for immediate postpartum Anemia is 11mg/dl which is the cutoff point of anemia after one week of postpartum, similarly, environmental factors like barefoot cause of anemia due to

hookworm were not included and a previous study conducted in a single facility but this study is conducted in ten Hospitals. The majority of this zone is highland in its topography. Therefore, assessing the burden of disease as well as risk factors of immediate postpartum anemia is very crucial especially in the study area and in the country at large.

## Methodology

### Study setting and population

An institutional-based cross-sectional study was conducted in East Gojjam zone Hospitals of Amhara regional state from October 20 to November 20, 2020 GC. East Gojjam zone is one of the 3rd most populous zones in the Region. It is found at 37.8087693 longitudes; 10.3287484 latitudes with 807meter minimum and 4236 meters elevation· It covered a total area of 14,009.74 square Kms. Based on the 2007 National Census conducted by the Central Statistical Agency of Ethiopia (CSAE), projection, and East Gojjam zone administration office report; this zone in 2019 has an estimated 2,719,118 population, of whom 1,335,123 are females, and 1,383,995 are males and the total female reproductive age groups are 91,634. It has 19 districts and 5 town administrations, which have 392 administrative kebeles and 40 urban kebeles. Debre Markos is the capital town of the East Gojjam zone, located 265 Km from Bahir Dar and 299 Km from Addis Ababa. This Zone has one comprehensive specialized Hospital, one General Hospital, 8 Primary Hospitals, 104 Health Centers, and 406 Health Posts [13]. There are about 10 hospitals in the study area as we stated above and a total of 2090 deliveries in ten Hospitals per month. Debre Markos Comprehensive Specialized hospital accounts 505, Mota General hospital 287, Lumamie Primary Hospital 249, Bibugn primary Hospital 186, Shebel Berenta primary hospital 183, Merto-lemariam primary hospital 160, Yejube primary hospital 160, Dejen Primary hospital 132, Bichena Primary hospital and Debre- work primary hospital 112 per month. This zone also has 22 seniors, 160 midwives, and 101 general practitioners in gynecology and obstetrics-related area of service [14].

### Sample size and sampling procedure

A total of 467 sample size was calculated by using Epi-Info stat Calc version 7.2 population survey by taking assumption of population size >10,000,95% confidence level, prevalence of immediate post-partum anemia 24.3% from previous study [15], margin of error 5%,design effect 1.5, 10% non-response rate. The sample size for the factor /the second objective was calculated by using Epi-Info statclc by taking five associated factors from previous studies and the greatest sample size among the five was 272 which is less than the sample size from the first objective then we took the sample size of the first objective.

A multistage sampling technique was employed to select the hospitals and the study participants with the assumption of homogeneity of the service in East Gojjam Zone hospitals. Five out of the ten were selected by simple random sampling technique using the lottery method. Bichena Primary hospital, Lumamie Primary Hospital, Shebel Berenta primary hospital, Merto-lemariam primary hospital, and Motta general hospital. The study participants were allocated to the proportion of client flow in each Hospital and the participants were selected by using a systematic random sampling technique every 2 case intervals before the mothers were discharged. $K^{th}$- interval, $K = \frac{N}{n,}$ where: N = Total population immediate post-partum at the selected five Hospitals, n = Total sample size, then k = 1,065/467 = approximately every 2 mothers will take but, calculating K-value for each hospital was necessary by using $K1 = \frac{N1}{/n1}$, $K2 = \frac{N2}{/n2}$. . . (K = 2.27, 2.24, 2.25 . . .. approximately every 2 immediate postpartum women was taken to each hospital after the first case selecting randomly between 1 and K).

## Study variables

**Dependent variable.** Immediate postpartum anemia.

**Independent variables. Socio-demographic factors** (age, residence, religion, ethnicity, marital status, educational status, occupation, and family income).

**Obstetrics history-related variables** (parity, multiple pregnancies, ANC follow-up, birth interval, previous abortion history, history of previous anemia, mode of delivery, tear, episiotomy, and neonatal birth weight).

**Dietary and micronutrient utilization variables** (iron-folic acid intake and mid-upper arm circumference).

**Environmental-related variables** (distance from a health facility, shoe-wearing status, and availability of toilet facility).

**Co-existing disease-related variables** (malaria, HIV/AIDS, tuberculosis).

**Operational definitions. Immediate postpartum period:** the time just after the child's birth by any mode of delivery via spontaneous vertex delivery, instrumental and cesarean delivery to the first 24 hours [16].

**Immediate postpartum anemia (IPA):** when the hemoglobin level is less or equal to 10g/dl in the immediate postpartum period, within 24 hours of post-delivery.

**Good adherence to IFA:** Supplementation means women who had taken iron folate supplements ≥90 days during the most recent pregnancy [17].

**Distance from health facility:** If the living house of the individual is greater than two hours walking from the health facility is considered as far from a health facility.

**MUAC:** If the individual postpartum woman has less than 23 cm of arm circumferences considered a lower MUAC in this study.

**Availability of toilet:** If an individual has any type of toilet in their compound or nearby their compound that is not used commonly consider as availability of toilet facility.

**Barefoot walking:** If people do not wear any types of shoes in their life considered as barefoot in this study.

## Data collection procedures and instrumentations

English version of the data collection interviewer-administered questionnaire, which was adapted from various kinds of literature [15, 18, 19] was used to collect the data from study participants and client's medical records from October 20-November 20,2020. It has five parts, socio-demographic characteristics, obstetrics history related, dietary and micronutrient utilization, Co-existing disease-related, and environmental-related characteristics. Five non-employed diploma midwives collected the data and two additional BSC midwives supervised the data collection process. Those data collectors and supervisors were recruited randomly by lottery method among non-employees. Data collection was done by these midwives who were not employed to minimize social desirability bias, one data collector for each hospital and two supervisors for all were assigned.

The data collectors took the lab request to the laboratory department for determination of the hemoglobin level after obtaining verbal informed consent eight hours after delivery then data collectors went to the laboratory and brought the EDTA tube and syringe with needle back to the mother then after 1 ml of venous blood was drowned from participants using aseptic technique by data collectors then taken to laboratory back for hemoglobin determination by using automated hematology analyzer Mindray done by laboratory technologies working at each hospital and determined. Data on IFA was collected by interviewing the woman for how long she took the supplement daily for a minimum of ninety days and MUAC was measured by tape meters measurement on a non-dominant hand following delivery, likewise, data on Co-existing disease was collected by reviewing her chart for investigations and interviewing her about known co-existing disease.

### Data quality control

To assure the data quality high emphasis was given to designing data collection tools. The pre-test was conducted on 5% of the sample size at Fenot Selam general hospital and necessary corrections on the instrument were employed accordingly. the one-day training was provided for data collectors and supervisors regarding the objectives of the study, data collection methods, the significance of the study, data collection tool, ethical considerations, and way of abstracting necessary information from the client's medical records and themselves. During data collection, assigned supervisors visited and supervised the data collection process and checked the completeness of the filled questionnaires.

### Data processing and analysis

All collected questionnaires were rechecked for completeness and coded; then, the data were entered and cleaned using Epi data version 4.2software and exported to SPSS version 25, for further analysis. Descriptive Statistics like frequency and cross-tabulation were carried out to characterize the study populations using socio-demographic and other variables.

Bivariable logistic regression was employed to identify the association between dependent and independent variables, those variables having a p-value of <0.25 in the Bivariable analysis were fitted into multivariable logistic regression analysis with backward like hood ratio to control the effects of confounding factors. Ninety-five percent confidence interval of odds ratios was computed and a p-value of less than 0.05 was considered to declare the statistical significance. The assumption of the binary logistic regression model was checked by using the Hosmer and Lemeshow test of goodness of fit test. Tables and graphical presentations were used to present the findings of the study.

### Ethical consideration

An ethical clearance letter was obtained from the Ethical Review Committee (ERC) of Debre Markos University College of Health Science with a reference number of HSC/R/C/Scr/Co/34/ 11/13. Upon bearing ethical review; the administration of each Hospital provided us permission to access the sample from the clients. The purpose of the study was explained and informed consent was obtained from each study participant. Privacy and confidentiality of all information were kept by coding throughout the research work.

## Result

### Socio-demographic characteristics

A total of 467 immediate postpartum women were included in the study, with a response rate of 100%. The age of respondents with immediate post-partum anemia ranged from 18 to 46 years, with a mean age of 28.19 years and SD of 6.3. The majority of 420 (89.9%) of the women were Orthodox religious followers and 453 (97%) were Amhara by ethnicity. About 443 (94.9%) mothers were married and more than half 250 (53.5%) of the mothers were urban in residency. About 146 (31.3%) of the respondents were unable to read and write and more than one-third 163 (34.9%) of the respondents were farmers. Among the respondents, 192 (41.1%) had a monthly family income of greater than 5,000 Ethiopian birrs (Table 1).

### Obstetrics characteristics

Among 467 study participants, 238(52%) were multipara. More than half 322 (69% of the mothers had term pregnancy. Most of the participants 428 (91.6%) had at least one ANC follow-up during their current pregnancy. The majority of the respondents 359 (76.9%) had no

**Table 1. Socio-demographic related characteristics of immediate postpartum women in East Gojjam zone hospital Northwest Ethiopia 2020 (n = 467).**

| Variables | Categories | Frequency (N) | Percentages (%) |
|---|---|---|---|
| Ages of the mother | ≤19 | 24 | 5.1 |
| | 20–34 | 354 | 75.8 |
| | >34 | 89 | 19.1 |
| Religion | Orthodox | 420 | 89.9 |
| | Muslim | 40 | 8.6 |
| | Protestant | 7 | 1.5 |
| Ethnicity | Amhara | 453 | 97.0 |
| | Oromo | 9 | 1.9 |
| | Tigray | 5 | 1.1 |
| Marital status | Married | 443 | 94.9 |
| | Unmarried | 24 | 5.1 |
| Residency | Rural | 217 | 46.5 |
| | Urban | 250 | 53.5 |
| Maternal education | Unable to read & write | 146 | 31.3 |
| | Able to read and write | 82 | 17.6 |
| | Primary | 111 | 23.6 |
| | Secondary | 39 | 8.4 |
| | Diploma and above | 89 | 19.1 |
| Maternal occupation | Farmer | 163 | 34.9 |
| | Merchant | 118 | 25.3 |
| | Housewife | 100 | 21.4 |
| | Gov't employee | 84 | 18.0 |
| | Others | 2 | 0.4 |
| Estimated monthly income | <1000 | 32 | 6.9 |
| | 1000–3000 | 132 | 28.3 |
| | 3001–5000 | 111 | 23.8 |
| | ≥5001 | 192 | 41.1 |

history of abortion. Among all 74 (15.8%) respondents had a history of Interpregnancy interval of fewer than two years. The majority of the respondents 391(83.7%) had no history of multiple pregnancies. Of the total study participants, 75 (16.1) had an antepartum hemorrhage in the last pregnancy. More than half 319 (68.3%) of the study participants gave birth through spontaneous vaginal delivery (SVD) (Table 2).

## Dietary and micronutrient characteristics

Of the study participants, 371 (79.4%) were start IFA during their recent pregnancy. Among these mothers who took IFA, 68.3% had poor adherence and IFA supplied participants 212 (45.4%) drank hot when they took iron. More than half of 263 (56.3%) ate three or above three times per day during their pregnancy time. Near one-third 147 (31.5%) of the mothers, mid-upper arm circumference was lower than 23cm (Table 3).

## Environmental characteristics and co-existing infections

Among all study participants, 128 (27.4%) had a co-existing infection (UTI, malaria, and tuberculosis). Thirty-six (7.7%) had a history of HIV/AIDS (Fig 1). The majority of the respondents 398 (85.2%) had a toilet facility and two-hundred eight-five of the women had wearing shoes always in their daily activities they are always using a shoe. Near to one-3rd of the respondents living in houses were far from the health facility less than two hours by walking.

**Table 2. Obstetrics characteristics related variables of immediate postpartum women in East Gojjam zone hospitals Northwest Ethiopia 2020 (n = 467).**

| Variables | Categories | Frequency (N) | Percentages (%) |
|---|---|---|---|
| Parity | Prime para | 229 | 49 |
| | Multipara | 238 | 51 |
| Gestational age | Preterm pregnancy | 111 | 23.7 |
| | Term pregnancy | 322 | 69 |
| | Post-term pregnancy | 34 | 7.3 |
| ANC follow-up | Yes | 428 | 91.6 |
| | No | 39 | 8.4 |
| Birth interval | <2 years | 74 | 15.8 |
| | ≥2 years | 169 | 36.2 |
| Antepartum hemorrhage | Yes | 75 | 16.1 |
| | No | 392 | 83.9 |
| Multiple pregnancies | Yes | 76 | 16.3 |
| | No | 391 | 83.7 |
| History of abortion | Yes | 108 | 23.1 |
| | No | 359 | 76.9 |
| Mode of delivery | SVD | 319 | 68.3 |
| | IAVD | 34 | 7.3 |
| | C/S | 114 | 24.4 |
| Episiotomy | Yes | 61 | 13.1 |
| | No | 292 | 62.5 |
| Perineal Tear | Yes | 99 | 21.2 |
| | No | 254 | 54.4 |
| Weight of newborn in grams | <2500 | 64 | 13.7 |
| | 2500–3999 | 382 | 81.8 |
| | ≥ 4000 | 21 | 4.5 |

CS = cesarean section, IAVD = Instrumental assisted vaginal delivery, SVD = spontaneous vertex delivery

**Table 3. Dietary and micronutrient uptake characteristics of immediate postpartum women in East Gojjam zone hospitals Northwest Ethiopia 2020 (n = 467).**

| Variables | Categories | Frequency (N) | Percent (%) |
|---|---|---|---|
| IFA tablet taken during pregnancy | Yes | 371 | 79.4 |
| | No | 96 | 20.6 |
| GA when IFA started | <16 weeks | 169 | 36.2 |
| | 20–24 weeks | 136 | 29.1 |
| | 26–30 weeks | 54 | 11.6 |
| | 30–34 weeks | 12 | 2.6 |
| Adherence to IFA supplementation | Good | 52 | 11.1 |
| | Poor | 319 | 68.3 |
| Hot drink when taking IFA | Yes | 212 | 45.4 |
| | No | 159 | 34.0 |
| Frequency of meals per day | ≤3 | 263 | 56.3 |
| | >3 | 204 | 43.7 |
| MUAC in centimeters | <23 | 147 | 31.5 |
| | ≥23 | 320 | 68.5 |

IFA = Irion-folic acid, GA = gestational age, MUAC = Mid upper arm circumference

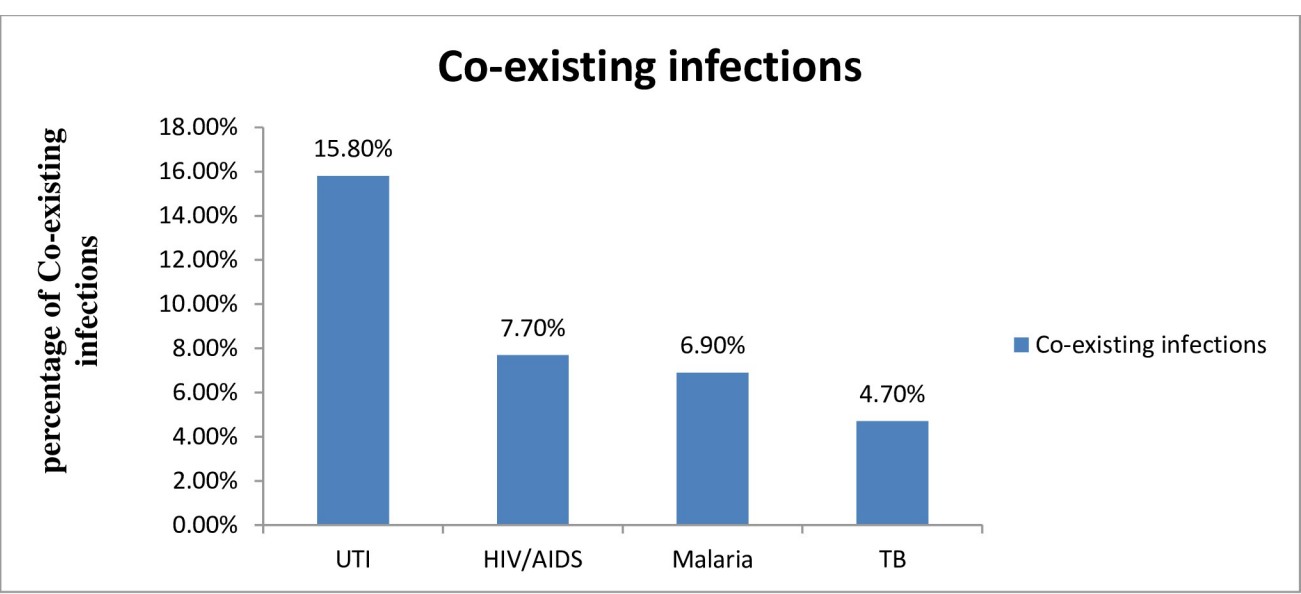

**Fig 1. Magnitude of co-existing infection among immediate postpartum women in the East Gojjam zone hospitals Northwest Ethiopia 2020.**

### The magnitude of immediate postpartum anemia

The overall magnitude of immediate postpartum anemia among mothers who gave birth in East Gojjam Zone Hospitals was found to be 101 (21.63%) 95% CI: [18.12%, 25.11%]).

### Factors associated with immediate postpartum anemia

In the Bivariable logistic regression analysis: residency, ANC follow-up, APH, mode of delivery, IFA intake during pregnancy, estimating distance from a health facility, availability of toilet facility, shoes wearing status, MUAC, and birth status of the newborns was associated with immediate postpartum Anemia at p-value less than 0.25.

As it is presented on the regression table not having ANC follow-up ((AOR = 2.92, 95%CI (1.20, 7.06)), having APH ((AOR = 4.51, 95% CI (2.42, 8.37)), instrumental assisted delivery ((AOR = 2.72, 95%CI (1.08, 6.78)), having C/S (AOR = 1.86, 95% CI (1.02, 3.35)) (MUAC) <23 cm(AOR = 5.75,95%CI (3.38, 9.79)), never used shoes((AOR = 2.60,95%CI (1.10, 6.14)), were significantly associated with immediate post-partum anemia in multivariable logistic regression model at p-value less than 0.05(Table 4).

### Discussion

A total of 101(21.6%,95%CI (18.1%, 25.1%) women experienced immediate postpartum anemia which is in line with the study done in the Amhara region, Northwest Ethiopia 24.3% [20], Tigray region Northwest Ethiopia 24.2% [21], Germany 25% [22], North Carolina, USA 19.1% [23]. The possible explanation might be increased ANC follow-up in those countries, which in turn, increases awareness of the women on how to minimize the development of immediate post-partum anemia, could be similar socio-demography health care delivery service, healthcare-seeking behavior of the women.

The finding of this study is lower than the study done in Jimma Zone, South Western Ethiopia 28.7% [24], in Mbarara regional referral hospital, Uganda 29.9% [25], in 12 US states of America 27% [26], India 26.5% [27]. This difference might be due to different scholars using

**Table 4. Bivariable and multivariable logistic regression analysis of factors associated with immediate postpartum anemia in East Gojjam zone hospitals Northwest Ethiopia 2020 (n = 467).**

| Variables | | Anemia | | p-value | COR 95%CI | AOR 95%CI |
|---|---|---|---|---|---|---|
| | | YES | NO | | | |
| Residence | Rural | 59 | 158 | 0.276 | 1.85 (1.83, 2.890) | 0.68 (0.340, 1.361) |
| | Urban | 42 | 208 | | I | I |
| ANC follow-up | No | 20 | 19 | 0.018 | 4.51 (2.301, 8.838) | 2.92 (1.203,7.066) * |
| | Yes | 81 | 347 | | I | I |
| Antepartum hemorrhage | Yes | 42 | 33 | 0.001 | 7.18 (4.214, 3.603) | 4.51 (2.427, 8.374) * |
| | No | 59 | 333 | | I | I |
| Mode of delivery | IAVD | 16 | 18 | 0.033 | 4.67 (2.235, 9.761) | 2.72 (1.086,6.787) * |
| | C/S | 34 | 80 | | 2.23 (1.354, 3.685) | 1.86 (1.029,3.350) * |
| | SVD | 51 | 268 | 0.040 | I | I |
| IFA intake during pregnancy | No | 32 | 64 | 0.692 | 2.19 (1.329, 3.603) | 1.17 (0.546, 2.491) |
| | Yes | 69 | 302 | | I | I |
| Estimated distance from health >2 facility in hours | | 41 | 81 | 0.309 | 2.40(1.506, 3.838) | 0.73 (0.391, 1.346) |
| ≤2 | | 60 | 285 | | I | I |
| Availability of toilet facility | No | 32 | 37 | 0.476 | 4.12 (2.404, 7.074) | 1.34 (0.602, 2.962) |
| | Yes | 69 | 329 | | I | I |
| Shoes wearing status | Sometimes | 40 | 99 | 0.346 | 2.30 (1.416, 3.752) | 1.34 (0.728, 2.477) |
| | Never | 17 | 16 | 0.029 | 6.06 (2.851, 12.884) | 2.60 (1.103, 6.149) * |
| | Always | 44 | 251 | | I | I |
| MUAC | < 23 | 67 | 80 | 0.001 | 7.04 (4.353, 11.401) | 5.75 (3.382, 9.790) * |
| | ≥23 | 34 | 286 | | I | I |
| Birth status of newborns who died | | 15 | 21 | 0.162 | 2.87 (1.418, 5.790) | 1.94 (0.767, 4.915) |
| Alive | | 86 | 345 | | I | I |

Note I = reference, * = significantly associated variables.

different cutoff values for hemoglobin measurements to determine postpartum anemia. Another possible reason might be mothers who develop anemia during their pregnancy were not included in this study.

The finding of this study is higher than those of the study done in Ghana 16% [28] and Burkina Faso 15.5% [29]. This discrepancy might be because the above study includes postpartum near to six-month postpartum period this gives time to recover from postpartum anemia.

Those mothers who had no ANC follow-up were three ((AOR = 2.92; 95%CI (1.20, 7.06)) times more likely to develop immediate postpartum anemia than those who had ANC follow-up. This is supported by the study done in Jimma Zone, South Western Ethiopia [24], Tigray region North West, Ethiopia [21], and Ethiopian demographic health survey [9]. The possible explanation could be women who had not had ANC follow-up not taken iron, might not be given anthelmintic for deworming of hookworm, and early identification of risk factors for postpartum anemia so that they are highly exposed to immediate postpartum anemia.

The odds of having immediate postpartum anemia among mothers diagnosed with APH were four ((AOR = 4.51; 95% CI (2.42, 8.37)) times more likely as compared to their counterparts. This is supported by the study conducted in the Amhara region, Northwest Ethiopia [20], and Germany [22]. The possible explanation might be due to the loss of iron stored during pregnancy and blood loss during delivery due to Antepartum hemorrhage. Those women who gave birth by instrumental assisted mode of delivery were three ((AOR = 2.72; 95%CI (1.08, 6.78)) times more likely to develop immediate postpartum anemia as compared to those

who gave birth through spontaneous vaginal delivery respectively. This result agreed with the studies done in the Amhara region, Northwest Ethiopia, and Spain [20, 30]. The possible explanation might be since instrumental assisted vaginal delivery increases the risk of episiotomy, spontaneous perineal or cervical tear, and this tear may be also extended to the uterus facilitating bleeding. Clinicians are usually misdiagnosing the tears and repairing after mothers bleed a lot.

The odds of getting immediate post-partum anemia among mothers who gave birth by cesarean delivery were two (AOR = 1.86; 95% CI (1.02, 3.35)) times more likely as compared to spontaneous vaginal delivery. The possible explanation could be in the case of cesarean deliveries there is the probability of inadvertent injury of the anterior abdominal wall and uterine vessels and a high possibility of extension to the fundal region of the uterus and vaginal wall.

Those women who had Mid upper arm circumference (MUAC) <23 cm were six (AOR = 5.75;95%CI (3.38, 9.79)) times more likely to develop immediate postpartum anemia than their counterparts. This finding is supported by the study done Amhara region Northwest, Ethiopia, Jemma, and Tanzania [20, 24, 31] respectively. The reason might be iron deficiency anemia usually related to micronutrient deficiency.

MUAC measurement <23 cm indicates that poor muscle mass lacks adequate energy intake and Hemoglobin concentration and maternal MUAC had a linear relationship.

Those women who never and sometimes used shoes were three and 1.34 ((AOR = 2.60;95% CI (1.10, 6.14)) and (AOR = 1.34;95% CI (0.72, 2.47)), times more likely to develop immediate postpartum anemia as compared to shoe users respectively. This finding is supported by a study done in Chennai, India [32]. The most possible explanation might be women who were walking barefoot are highly exposed to hookworm infestation which is one of the risk factors for postpartum anemia.

## Strength of the study

As the study is conducted from primary data and more of it is clinical imperative to indicate a clear magnitude or image of immediate post-partum anemia in the study area.

## Limitations

Since the study is cross-sectional causality relationship may not be established and behavioral factors were not assessed. Because the interview was about the past nine months 'recall bias was one of the challenges.

## Conclusion

This study indicates that immediate postpartum anemia is a moderate public health problem in the study area. A more careful strategy is ideal to increase ANC follow-up that sticks to national guideline contact schedule, safe reduction of instrumental and cesarean deliveries to the minimum, quick and timely linkage and treatment of malnourished pregnant mothers to the center where they get adequate health care services, along with high index of suspicion in mothers diagnosed with APH, wise and vigilant advice on consistent use of the shoe for pregnant mothers are recommended to tackle the burden of immediate post-partum anemia.

## Supporting information

**S1 Data.**
(SAV)

**S1 File.**
(ZIP)

## Acknowledgments

We would like to express our deepest and heartfelt appreciation to Debre Markos University for providing the opportunity to conduct this research work. We are also happy to say thanks to East Gojjam Zone Hospital's administration and community for their kindness and their valuable information to do this work. Last but not least, we express our gratitude to study participants, data collectors, and supervisors for their contribution to this work.

## Author Contributions

**Conceptualization:** Getachew Altaseb Agmassie, Genet Degu Alamneh, Moges Wubie Ayicheh.

**Investigation:** Getachew Altaseb Agmassie, Genet Degu Alamneh, Moges Wubie Ayicheh, Worku Taye Getahun, Aysheshim Asnake Abneh.

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
