## [Decision Letter · Decision Letter 0]

15 Nov 2021

PONE-D-21-22500Magnitude and associated factors of immediate postpartum anemia among women who gave birth in east Gojjam zone hospitals, northwest- Ethiopia, 2020PLOS ONE

Dear Dr. Getahun,

Thank you for submitting your manuscript to PLOS ONE. After careful consideration, we feel that it has merit but does not fully meet PLOS ONE’s publication criteria as it currently stands. Therefore, we invite you to submit a revised version of the manuscript that addresses the points raised during the review process.

We look forward to receiving your revised manuscript.

Kind regards,

Desalegn Admassu Ayana, Ph.D

Academic Editor

PLOS ONE

Journal Requirements:

Whilst you may use any professional scientific editing service of your choice, PLOS has partnered with both American Journal Experts (AJE) and Editage to provide discounted services to PLOS authors. Both organizations have experience helping authors meet PLOS guidelines and can provide language editing, translation, manuscript formatting, and figure formatting to ensure your manuscript meets our submission guidelines. To take advantage of our partnership with AJE, visit the AJE website (http://aje.com/go/plos) for a 15% discount off AJE services. To take advantage of our partnership with Editage, visit the Editage website (www.editage.com) and enter referral code PLOSEDIT for a 15% discount off Editage services.  If the PLOS editorial team finds any language issues in text that either AJE or Editage has edited, the service provider will re-edit the text for free.

3. Thank you for stating the following in the Acknowledgments/ Funding Section of your manuscript: 

The financial support for this research project was obtained from Debre Markos University. The funder had no role in the design of the study and collection, analysis, and interpretation of data and in writing the manuscript. 

5.  Thank you for submitting the above manuscript to PLOS ONE. During our internal evaluation of the manuscript, we found significant text overlap between your submission and the following previously published works, some of which you are an author.

http://repository-tnmgrmu.ac.in/8960/1/201500418samykkhan.pdf

https://www.omicsonline.org/open-access/postpartum-anemia--still-a-major-problem-on-a-global-scale-2376-127X-1000e122.php?aid=61457

https://www.hindawi.com/journals/anemia/2020/8979740/

Please revise the manuscript to rephrase the duplicated text, cite your sources, and provide details as to how the current manuscript advances on previous work. Please note that further consideration is dependent on the submission of a manuscript that addresses these concerns about the overlap in text with published work.

Reviewers' comments:

Reviewer's Responses to Questions

**Comments to the Author**

1. Is the manuscript technically sound, and do the data support the conclusions?

Reviewer #1: Yes

Reviewer #2: Partly

2. Has the statistical analysis been performed appropriately and rigorously? 

Reviewer #1: I Don't Know

Reviewer #2: Yes

3. Have the authors made all data underlying the findings in their manuscript fully available?

Reviewer #1: Yes

Reviewer #2: Yes

4. Is the manuscript presented in an intelligible fashion and written in standard English?

Reviewer #1: No

Reviewer #2: No

5. Review Comments to the Author

Reviewer #1: Specific comments

Abstract

Background: line 20, “less or equal to 10 g/dl in”, but according to World Health Organization (WHO), anemia is defined as hemoglobin (Hb) levels <12.0 g/dL in women. Do you have explanation for the different definition?

Conclusion: line 41, how much is “a moderate public health problem”? 21.6? What is the implication of your finding?

Introduction: why do you classified introduction in two background and statement of the problems? Further, the so called background was not well articulated.

Line 76, “anemia in the developed country ranges from 10-30% and in developing country 50-80%”, it is preferable if data from low and middle income countries used instead of developing countries. What is the global burden? The citation are too old to put current image of the problems (for instance, reference 10, 11, 12). Please update other references accordingly.

The Excerpt you cited have no the idea you mentioned in the documents i.e., “line 84-89” or reference 16 or “https://pubmed.ncbi.nlm.nih.gov/27227230/”

Line 92-95, are those consequence of anemia serious enough to design this study?

Line 114, “assessing the burden of disease” how health facility-based study reveal burden?

What did your study add to existing knowledge? In addition to what Abebaw et al, 2020 were explored.

Sample size and sampling procedure: was sample size calculated for factors? If no why?

Why multistage? How many stages were there? Why design effect of 1.5?

Line 139, “every 2 case intervals” how did you find 2? Why not 3?

Data collection procedures and instrumentations: cite at least few references from where you adopted the questionnaire.

The variables measurement need major revision. Laboratory procedure also need further explanation. How blood sample was taken? Who take the blood sample? How blood sample was transported? Etc.

Result: line 184-185, age explained in three ways. Avoid redundancy.

Bivariable and multivariable logistic regression analysis: where are explanation of multivariable analysis?

Discussion: the discussion begin with summary findings. If your study is in line with “Amhara region, Northwest Ethiopia 24.3% (23), Tigray region Northwest Ethiopia 24.2% (24), Germany 25%(25), North Carolina, USA 19.1%(9)” what it add to existing knowledge?

“different scholars used different cutoff values for hemoglobin measurements to determine postpartum anemia.” Why you compare with already different study? But you didn’t mention “Another possible reason may be mothers who develop anemia during their pregnancy were not included in this study.” in your exclusion?

Your discussion should look different from result.

The discussion need through writing, highlighting the implication for policy.

The references style was not plos one style and need correction and updating.

What are the strength and limitation of your study?

Conclusion: What are the policy implication of the documents?

Reviewer #2: My comment and question as follows

The Authors touched magnitude and associated factors of immediate postpartum anemia among women who gave birth. The authors rise interesting issue which need further researching to generate evidence that support for planner to development of strategic plan and health care provider better intervention in preventing maternal mortality, and effect on their children

General comment after delivery when the data collected (blood specimen)? No time limit and question inclusion and exclusion criteria was not mentioned

Is there a difference in cut of point to define anemia after delivery? Based on the duration? If so how managed?

Specific comment and question

Abstract

Line 21: Background: add a sentences that indicate the gap, gap and severity of the problem are different

Result

Line 36-39 avoid upper case unnecessary used

Line 35-39: Use standard Caption problem = (21.6% = [95%, CI (18.1, 25.1)]) others

Conclusion

Line 41: Immediate postpartum anemia is a moderate public health problem, not appropriate because 21.6% anemic mother

Line 42: “affected” change to associated by not having ANC follow up,…..

Introduction: need further synthesis, need to full fil the format for manuscript and need further work

Method

Good if study setting focus on delivery services

Study participant after delivery not clearly mentioned because the cut of point for anemia different

How data collection of dietary and micronutrient utilization conducted? How co-existing infection collected and measured?

Further description of the test used for anemia diagnose its advantage, sensitivity and reliability quality control, the procedure data collected…

Result

Thirty-six (7.7%) were had a history of HIV/AIDS. not clear is prevalence

Near to one-3rd of the respondents living houses 214 were far from health facility less than two hours by walking/greater than?

Description of Bivariable and multivariable logistic regression analysis interpretation required

Discussion

Not strongly discussed on implication and scientific background of the finding

6. PLOS authors have the option to publish the peer review history of their article (what does this mean?). If published, this will include your full peer review and any attached files.

Reviewer #1: **Yes: **Kasiye Shiferaw

Reviewer #2: No

---

## [Author Response · Author response to Decision Letter 0]

30 Dec 2021

Response Reviewers’ comments:

 Is the manuscript technically sounds, and do the data support the conclusions?

Response: the manuscript is technically sounds and the data drawn based on the data presented 

 Has the statistical analysis been performed appropriately and rigorously?

Response: statistical analysis been performed with appropriate model of regression and rigorously 

 Have the authors made all data underlying the findings in their manuscript fully available?

Response: all data are available fully in the manuscript 

 Is the manuscript presented in an intelligible fashion and written in Standard English?

Response: We reviewed the entire document thoroughly for the English Language usage, spelling, grammar in an intelligible fashion and the necessary changes were made based on the comment given 

Background: line 20, “less or equal to 10 g/dl in”, but according to World Health Organization (WHO), anemia is defined as hemoglobin (Hb) levels <12.0 g/dL in women. Do you have explanation for the different definition?

Response: according to WHO definition of Anemia in a women is Hgb levels <12.0g/dl, but during pregnancy there is physiologic change that means there is an increase both in red blood cell and plasma level despite of this the increase in red blood cell by about 33% and plasma level by about 50% is not proportional that leads to hem dilution and drop of the Hgb level during pregnancy, so that is the reason the cutoff point Hgb <=10g/dl to define anemia and it doesn’t regain its non-pregnant state Hgb level 

 How much is “a moderate public health problem”? 21.6? What is the implication of your finding?

Response: based on its public health importance postpartum anemia can be classified as normal mild, moderate and severe according to its prevalence. If the prevalence is ≤4.9% it is normal no public health problem, 5-19.9% is mild public health problem, 20-39.9% is moderate public health problem, ≥40% severe public health problem, so the prevalence of the is finding is 21.6% and put it under moderate public health problem as per to the classification needs attention to avert because of it public health problem. 

Why do you classified introduction in two background and statement of the problems? Further, the so called background was not well articulated.

Response: we will revise and make the two sections of the introduction 

What did your study add to existing knowledge? In addition to what Abebaw et al, 2020 were explored

Response: environmental factors were not assessed from previous studies and for immediate post-partum anemia Hgb measurement is taken from cutoff point 10-10.5g/dl in WHO recommendation but in the previous studies used the cut point of 11g/dl at one week of postpartum other than immediate postpartum that affects the overall image of the finding. 

Sample size and sampling procedure: was sample size calculated for factors? If no why?

Response; sample size for the factor was calculated by using Epi-Info statclc 

Why multistage? How many stages were there? Why design effect of 1.5?

Response: there are ten Hospitals in the study area of them five were selected randomly by using lottery method then study participants were taken from five hospital which means we go dawn from ten to five the proportional allocation was used and has two stage, based on the sage we can use design effect of two but due to its cost wise we used design effect 1.5 

“every 2 case intervals” how did you find 2? Why not 3?

Response: K- interval, K= N/n , where: N = Total population immediate post-partum at the selected five Hospitals , n=Total sample size, then k=1,065/467 = approximately every 2 mothers will take but, calculating K-value for each hospital was necessary by using K1=N1/n1, K2=N2/n2……….(K= 2.27, 2.24 ,2.25 …. approximately every 2 immediate postpartum women was taking for each hospital after the first case selecting randomly between 1 and K). if we take 3 it becomes 1401 which is more than the Total population immediate post-partum at the selected five Hospitals due to this we took 2

If your study is in line with “Amhara region, Northwest Ethiopia 24.3% (23), Tigray region Northwest Ethiopia 24.2% (24), Germany 25%(25), North Carolina, USA 19.1%(9)” what it add to existing knowledge

Response: despite the fact that the magnitude of the study is in line with these previous studies environmental factors were not assessed from previous studies.

“different scholars used different cutoff values for hemoglobin measurements to determine postpartum anemia.” Why you compare with already different study? But you didn’t mention “Another possible reason may be mothers who develop anemia during their pregnancy were not included in this study.” in your exclusion?

Response: in our study mothers who were anemic in their ANC follow up were assessed and excluded but previous studies were not exclude mothers who were anemic during their ANC follow up that is we want to explain but not compare studies done on pregnant, mothers 

Your discussion should look different from result

Response: we revised the discussion session 

Reviewer #2: 

When the data collected (blood specimen)? No time limit and question inclusion and exclusion criteria was not mentioned Is there a difference in cut of point to define anemia after delivery? Based on the duration? If so how managed?

Response: the data were collected with 24 hours of post- delivery, yes the time matters after delivery to define post-partum anemia , based on WHO current recommendation for immediate post-partum /within 24 hours of anemia the cutoff point is 10g/dl whereas as at one week of post-partum 11g/dl at eight weeks of post-partum 12 g/dl is used to determine anemia after delivery and the management is based on the time and level of Hgb 

How data collection of dietary and micronutrient utilization conducted? How co-existing infection collected and measured?

Response: by using adapted questioners like Irion and folic acid intake during pregnancy, for how many times took iron and folic acid supplementation in this Pregnancy, Have you drink coffee and tea when taking IFA Average Frequency of meal per day and MUAC measurement Helminths infestation, known past medical disease

---

## [Decision Letter · Decision Letter 1]

7 Mar 2022

PONE-D-21-22500R1Magnitude and associated factors of immediate postpartum anemia among women who gave birth in east Gojjam zone hospitals, northwest- Ethiopia, 2020PLOS ONE

Dear Worku, 

Thank you for submitting your manuscript to PLOS ONE. After careful consideration, we feel that it has merit but does not fully meet PLOS ONE’s publication criteria as it currently stands. Therefore, we invite you to submit a revised version of the manuscript that addresses the points raised during the review process.

We look forward to receiving your revised manuscript.

Kind regards,

Desalegn Admassu Ayana, Ph.D

Academic Editor

PLOS ONE

Journal Requirements:

Additional Editor Comments (if provided):

All the results should be presented with data and the manuscript should be reviewed by native English language speaker.

Reviewers' comments:

Reviewer's Responses to Questions

**Comments to the Author**

1. If the authors have adequately addressed your comments raised in a previous round of review and you feel that this manuscript is now acceptable for publication, you may indicate that here to bypass the “Comments to the Author” section, enter your conflict of interest statement in the “Confidential to Editor” section, and submit your "Accept" recommendation.

Reviewer #1: (No Response)

Reviewer #2: (No Response)

2. Is the manuscript technically sound, and do the data support the conclusions?

Reviewer #1: Yes

Reviewer #2: Yes

3. Has the statistical analysis been performed appropriately and rigorously? 

Reviewer #1: Yes

Reviewer #2: Yes

4. Have the authors made all data underlying the findings in their manuscript fully available?

Reviewer #1: No

Reviewer #2: Yes

5. Is the manuscript presented in an intelligible fashion and written in standard English?

Reviewer #1: No

Reviewer #2: No

6. Review Comments to the Author

Reviewer #1: Reviewer comments

The papers seems improved, but there are a lot things that authors should consider to improve the manuscript before publishing it. The paper need thorough English language edition. Grammar, conjunctions and punctuation need correction. The gap of the study should be clearly stated in relation to what others failed to achieve.

Abstract

Line 35: delete the reference. Begin the sentence appropriately.

Line 35-39: Be consistent and make every numbers two decimal point.

Introduction

What other studies identified are explained insufficiently.

The authors failed to show the gap of the study. Although they claimed environmental factors were not assessed in the author response. What makes this study different from Abebaw et al., 2020 study?

Method section

Variables and measurement are absent. There are too many variables. Do you think your sample size is adequate?

How quality of laboratory test was maintained? Is automated blood analyzer cell dyne 1800 appropriate test? How about its sensitivity and specificity?

Result

Line 184: Table number is missing.

Line 199-202: How did you verified the women have toilet facility? What do you mean by wearing shoes always in their daily activities? I didn’t see any operational definition in this document.

Line 204-207: what is the difference between two sentences?

Line 208: rewrite the title.

In the tables some cell of the tables is very few, you better consider re-categorizing them. The table legend should include abbreviations. For instance SVD? IAVD?

Line 213-219: table 4 cited twice.

Discussion

Is it fair to compare Ethiopia to German, USA? You better compare with LMICs.

The discussion seems result section, please interpret and implicate the finding. You should avoid odds ratio in the discussion section.

Limitation: What other potential limitation do you encountered? What about recall and measurement bias?

Reviewer #2: The question I have raised and responded by the Authors need to be incorporated in the manuscript

1. a difference in cut of point to define anemia after delivery? Based on the duration? If so how managed? need to be incorporated in methodology part how practically data collected or measured

2. How data collection of dietary and micro nutrient utilization conducted? How co-existing

infection collected and measured? Only iron and folic acid supplementation data collected on the manuscript need to be inline with this, how co-existing infection data collected also need to be incorporated in manuscript

7. PLOS authors have the option to publish the peer review history of their article (what does this mean?). If published, this will include your full peer review and any attached files.

Reviewer #1: **Yes: **Kasiye Shiferaw Gemechu

Reviewer #2: No

---

## [Author Response · Author response to Decision Letter 1]

17 Apr 2022

Response to Reviewers’ comments 

Reviewer #1: 

1. The papers seem improved, but there are a lot of things that authors should consider to improve the manuscript before publishing it. The paper need thorough English language edition. Grammar, conjunctions and punctuation need correction. The gap of the study should be clearly stated in relation to what others failed to achieve.

Response: We reviewed the entire document thoroughly for the English Language usage, spelling, grammar, and the necessary changes were made.

-the sample size is adequate 

2. What other studies identified are explained insufficiently.

The authors failed to show the gap in the study. Although they claimed environmental factors were not assessed in the author's response. What makes this study different from Abebaw et al., 2020 study?

Response: We included the gap between other previous studies, in the revised manuscript of this study.

3. Variables and measurements are absent. There are too many variables. Do you think your sample size is adequate? How quality of the laboratory test was maintained? Is an automated blood analyzer cell dyne 1800 appropriate test? How about its sensitivity and specificity?

Response: sorry that we have mentioned automated blood analyzer cell dyne 1800 wrongly we used the mindray hematology analyzer in the revised manuscript.

- We reviewed the entire document thoroughly for the English Language usage, spelling, grammar in an intelligible fashion and the necessary changes were made based on the comment given

- We minimized the use of abbreviations in the table and, we added the list of

Acronyms we used throughout the manuscript.

Reviewer #2: 

The question I have raised and responded by the Authors need to be incorporated in the manuscript

1. A difference in cut of point to define anemia after delivery? Based on the duration? If so how managed? Need to be incorporated in methodology part how practically data collected or measured

Response: We included the cut off point to define anemia after delivery in the revised manuscript 

2. How data collection of dietary and micro nutrient utilization conducted? How co-existing infection collected and measured?

Response: we included the way how data were collected in the revised manuscript

---

## [Decision Letter · Decision Letter 2]

7 Jul 2022

PONE-D-21-22500R2Magnitude and associated factors of immediate postpartum anemia among women who gave birth in east Gojjam zone hospitals, northwest- Ethiopia, 2020PLOS ONE

Dear Dr. Getahun,

Thank you for submitting your manuscript to PLOS ONE. After careful consideration, we feel that it has merit but does not fully meet PLOS ONE’s publication criteria as it currently stands. Therefore, we invite you to submit a revised version of the manuscript that addresses the points raised during the review process.

Please carefully address the remaining concerns provided by the reviewer, and please ensure that the entire manuscript is thoroughly copy edited for grammar and usage as well as typographical errors. 

We look forward to receiving your revised manuscript.

Kind regards,

Vanessa Carels

Staff Editor

PLOS ONE

Reviewers' comments:

Reviewer's Responses to Questions

**Comments to the Author**

1. If the authors have adequately addressed your comments raised in a previous round of review and you feel that this manuscript is now acceptable for publication, you may indicate that here to bypass the “Comments to the Author” section, enter your conflict of interest statement in the “Confidential to Editor” section, and submit your "Accept" recommendation.

Reviewer #2: (No Response)

2. Is the manuscript technically sound, and do the data support the conclusions?

Reviewer #2: Yes

3. Has the statistical analysis been performed appropriately and rigorously? 

Reviewer #2: Yes

4. Have the authors made all data underlying the findings in their manuscript fully available?

Reviewer #2: Yes

5. Is the manuscript presented in an intelligible fashion and written in standard English?

Reviewer #2: No

6. Review Comments to the Author

Reviewer #2: Comment as reviewer

The author’s raises important topic contributing maternal morbidity and mortality, my general comment is there are many evidence in the country on the area. The evidence generated were already explored by previously conducted researchers in different part of the countries example Amhara region similar region and Tigray region the study lacks novelty and gap not well addressed.

Abstract background Gap not addressed

Introduction not focus on the study objectives PP anemia, the characteristics of the study area were not addressed. There are many related study authors have mentioned in Ethiopia, why study needed since Ethiopia context is similar

Method:

Sampling technique not clear there was a missed information, check it

How systematic random sampling method Data were used since PNC in Ethiopia is very low? When blood collected and hemoglobin measured not clarified, how anemia identified depending on the duration after delivery the standard not clarified

Co-existing disease-related variables mentioned (malaria, HIV/AIDS, tuberculosis)why important hook work infection not addressed since the authors mentioned bare foot?

What quality assurance measure taken for blood collection and determining hemoglobin level?

What measure taken for identified anemic women on ethical issue

Result part marital status n=465 why?, episiotomy, perinatal tear check all result

Table quality need modification

Among these mothers who took IFA, 68.3% had poor adherence can we say poor adherence by collecting retrospective data , what is the standard to measure adherence what data collection method authors have to use?

The authors collected hot drink when taking IFA, how data collected can we collect valid and reliable data for such specific information, need to put as limitation for such specific issues

Discussion

Discussion need further work on evidence based reasoning for the finding in comparison, addressing implication of the finding on the study participates, implementation at health facilities and scientific explanation need to be addressed

Language edition need further work

7. PLOS authors have the option to publish the peer review history of their article (what does this mean?). If published, this will include your full peer review and any attached files.

Reviewer #2: No

---

## [Author Response · Author response to Decision Letter 2]

21 Aug 2022

Response to Reviewers’ comments 

1. My general comment is there is much evidence in the country on the area. The evidence generated were already explored by previously conducted researchers in different part of the countries example Amhara region similar region and Tigray region the study lacks novelty and gap not well addressed.

Response: We included the gap between other previous studies, in the revised manuscript of this study and there is a difference in topography in this study area and previous studies despite being in the same region, and anemia is affected by geographic location.

2. Introduction not focus on the study objectives PP anemia, the characteristics of the study area were not addressed. There are many related study authors have mentioned in Ethiopia, why study needed since Ethiopia context is similar?

Response: We included the gap between other previous studies, in the revised manuscript of this study, and the study area is characterized in the revised manuscript.

3. Sampling technique is not clear there was missed information, check it how systematic random sampling method Data were used since PNC in Ethiopia is very low? When blood collected and hemoglobin measured not clarified, how anemia identified depending on the duration after delivery the standard not clarified Co-existing disease-related variables mentioned (malaria, HIV/AIDS, tuberculosis) why important hook work infection not addressed since the authors mentioned bare foot?

Response: sampling technique is stated in the revised manuscript; the data were collected before mothers had been discharged to home and collected eight hours after delivery, we considered that hook warm infestations are explained by the bare boot 

4. What quality assurance measure taken for blood collection and determining hemoglobin level?

Response: one-day training was given to data collectors and supervisors before they went to collect the data about ways of data collection and how to transfer to the laboratory 

5. What measure taken for identified anemic women on ethical issue

Response: After determination of hemoglobin therapeutic dose of iron was supplemented before discharge for all anemic women

6. Result part marital status n=465 why? Episiotomy, perinatal tear check all result

Table quality need modification 

Response: sorry it is editorial problem we checked and corrected down to the document, episiotomy and perineal tear were taken among vaginal deliveries only not cesarean deliveries 

7. Among these mothers who took IFA, 68.3% had poor adherence can we say poor adherence by collecting retrospective data , what is the standard to measure adherence what data collection method authors have to use?

Response: supplementation was extracted from their chart and adherence was taken from an interview of the mothers 

8. Language edition need further work

Response: We reviewed the entire document thoroughly for the English Language usage, spelling, and grammar in an intelligible fashion, and the necessary changes were made based on the comment given

---

## [Decision Letter · Decision Letter 3]

24 Feb 2023

Magnitude and associated factors of immediate postpartum anemia among women who gave birth in east Gojjam zone hospitals, northwest- Ethiopia, 2020

PONE-D-21-22500R3

Dear Dr. Worku Taye Getahun,

We’re pleased to inform you that your manuscript has been judged scientifically suitable for publication and will be formally accepted for publication once it meets all outstanding technical requirements.

Kind regards,

Malshani Lakshika Pathirathna, PhD

Academic Editor

PLOS ONE

Additional Editor Comments (optional):

Reviewers' comments:

Reviewer's Responses to Questions

**Comments to the Author**

1. If the authors have adequately addressed your comments raised in a previous round of review and you feel that this manuscript is now acceptable for publication, you may indicate that here to bypass the “Comments to the Author” section, enter your conflict of interest statement in the “Confidential to Editor” section, and submit your "Accept" recommendation.

Reviewer #3: All comments have been addressed

2. Is the manuscript technically sound, and do the data support the conclusions?

Reviewer #3: Yes

3. Has the statistical analysis been performed appropriately and rigorously? 

Reviewer #3: Yes

4. Have the authors made all data underlying the findings in their manuscript fully available?

Reviewer #3: Yes

5. Is the manuscript presented in an intelligible fashion and written in standard English?

Reviewer #3: Yes

6. Review Comments to the Author

Reviewer #3: General comment: Please correct the grammar, spacing and punctuation error before publication.

Make it clear Table (1)-Educational Status

Add foot Note for Table (2)-C/S, IAVD, SVD

7. PLOS authors have the option to publish the peer review history of their article (what does this mean?). If published, this will include your full peer review and any attached files.

Reviewer #3: **Yes: **Mulualem Silesh

---

## [Editor Report · Acceptance letter]

6 Mar 2023

PONE-D-21-22500R3 

The magnitude and associated factors of immediate postpartum anemia among women who gave birth in east Gojjam zone hospitals, northwest- Ethiopia, 2020 

Dear Dr. Getahun:

I'm pleased to inform you that your manuscript has been deemed suitable for publication in PLOS ONE. Congratulations! Your manuscript is now with our production department. 

Kind regards, 

on behalf of

Dr. Malshani Lakshika Pathirathna 

Academic Editor

PLOS ONE